# Prevalence, Infection, and Risk to Human Beings of *Toxocara canis* in Domestic Food-Producing Animals

**DOI:** 10.3390/vetsci11020083

**Published:** 2024-02-09

**Authors:** Jingyun Xu, Qian Han

**Affiliations:** 1Laboratory of Tropical Veterinary Medicine and Vector Biology, School of Life and Health Sciences, Hainan University, Haikou 570228, China; jingyunxu@hainanu.edu.cn; 2One Health Institute, Hainan University, Haikou 570228, China

**Keywords:** toxocariasis, zoonosis, epidemiology

## Abstract

**Simple Summary:**

*Toxocara canis* is a common parasite that resides in the intestinal tract of dogs. It can be transmitted through soil, water, or vegetables that are contaminated with infected eggs. However, another transmission is through the consumption of raw or undercooked meat from infected paratenic hosts. The paratenic host for *T. canis* includes various food-producing animals such as mammals and birds. Unfortunately, the infection has been largely overlooked by the general public. Therefore, the purpose of this review is to shed light on the parasitism, migration, and infection of *T. canis* in food-producing animals. We aim to raise public awareness about toxocariasis.

**Abstract:**

Toxocariasis is a significant food-borne zoonotic parasitic disease, and a range of birds and mammals are the paratenic hosts of *Toxocara canis*. The consumption of raw or undercooked meat and viscera of these paratenic hosts frequently leads to *T. canis* infection and the development of human toxocariasis. In this review, we will perform an analysis of relevant papers published in the National Center for Biotechnology Infrastructure database on the parasitism, migration, and infection of *T. canis* in chickens, pigeons, quail, pigs, cattle, sheep, and other food-producing animals, so as to make the public aware of the risk factors of human toxocariasis, improve the public’s understanding of *T. canis* infection, and provide evidence for targeted prevention and control measures.

## 1. Introduction

Toxocariasis is a prevalent zoonotic worm disease that holds significant global importance. Unfortunately, it is also classified as one of the most neglected zoonotic diseases [1]. The prevalence of the disease is higher in many developing countries, especially in countries and regions with diets of uncooked or raw meat [2,3,4,5]. The disease is considered to be a soil-borne parasitic disease, with food-borne transmission primarily associated with the consumption of raw or undercooked meat and internal organs from paratenic hosts, including birds [6,7,8,9]. Human infection with *Toxocara canis* is primarily categorized into three types of exposure: direct, nearly direct, and indirect. In the case of direct exposure, individuals come into direct contact with *T. canis*-infected dogs, which can result in an infection. Additionally, children can become infected by playing in parks that have soil contaminated with *T. canis* eggs. Moreover, apart from consuming paratenic hosts infected with *T. canis*, infection can also occur by consuming vegetables and water that are contaminated with *T. canis* eggs [10,11,12]. As the paratenic host of *T. canis*, the parasite cannot complete its life cycle in the human body, but its infective larvae can migrate to various organs and tissues through the circulatory system [13]. Although human toxocariasis is usually asymptomatic, the inflammatory response triggered by the larvae can lead to severe infection with liver, lung, eye, and nervous system complications [14,15].

According to the available epidemiological data, there may be significant differences in serum prevalence among different populations. For example, preschoolers and people living in rural areas or engaged in land-related occupations have a higher risk of toxocariasis [16,17]. The prevalence of anti-*T. canis* antibodies in random patient samples which were admitted to a children’s hospital in Brazil ranged from 31 to 39% [18]. And the human infection rate ranges from 11% to 52% [19]. Additionally, survey results indicated that people living in rural areas had minimal awareness of the risk of zoonotic diseases transmitted from raw meat. The comprehensive summary provided by Healy et al. [20] reveals that pigs, lambs, and chickens pose a significant risk for transmitting toxocariasis. Currently, cases of human toxocariasis have been reported due to the consumption of raw chicken liver [6,21], duck liver [22], sheep liver [23], and cattle liver [24]. However, the clinical and laboratory diagnosis of toxocariasis is relatively complex, making it difficult to determine the extent of infection in the population [17]. On the other hand, although human toxocariasis is associated with the consumption of animal-derived foods, the epidemiological investigation of these food-borne animals infected with *T. canis* is rarely reported. In this review, we will expand our investigation to a broader range of food-producing animals and focus on the infection, migration, and transmission of *T. canis* within these animals. By analyzing the results of *T. canis* experimental infections, epidemiological investigation data, and human toxocariasis cases due to consumption of raw or undercooked meat from food-producing animals, we will identify potential risk factors that contribute to human toxocariasis. This review aims to not only help establish preventive and control measures against toxocariasis, but also to raise public awareness about the potential risks associated with *T. canis* infection.

A comprehensive literature search was performed to gather relevant papers on the parasitism, migration, and infection of *T. canis* in various food-producing animals such as chickens, pigeons, quail, pigs, cattle, sheep, and others. The search was conducted in the National Center for Biotechnology Infrastructure (NCBI) database. No limitations were placed on the time frame or language of the publications. The search utilized the keywords “*Toxocara canis*” in combination with the animal species, including “chicken”, “pigeon”, “quail”, “pig”, “cattle”, and “sheep”. Each paper was meticulously reviewed to exclude irrelevant and duplicated studies. We first collected 56 papers, and according to the reviewing criteria, 52 papers were finally included in the analysis. These papers included experimental infections, epidemiological investigations, case reports, and similar studies. Papers that were not accessible in their entirety or did not align with the topic were excluded from the analysis.

## 2. Experimental Infection

### 2.1. Chicken

Broilers are one of the primary food commodities in the world. Traditionally, broilers are raised in semi-intensive or free-range systems in village chicken farms [25,26], and chickens are more susceptible to soil-borne parasite infections due to their ability to wander unrestrictedly in the farm in these systems. In addition, chickens are also considered to be indicators of soil-borne parasites, especially *Toxocara* spp. [27,28]. Studies have found that these infected chickens may release *T. canis* eggs into the environment 2 to 6 h post-infection (hpi), and the eggs in feces are still infective [29], which indicates the possibility of cross-infection between chickens. This also increases the risk of infection for breeders to some extent.

In 1987, Inoue et al. experimentally inoculated *T. canis* eggs into the gizzard of chickens and observed larvae in the liver [30]. In addition, they fed the livers from chickens infected with *T. canis* to mice, which led to the mice being successfully infected. This is the first study to investigate the possibility of chicken as a paratenic host for *T. canis*. Similarly, Dutra et al. successfully infected mice by feeding them livers from chickens infected with *T. canis* larvae (n = 50), further demonstrating that the *T. canis* larvae in the livers of chickens are still infectious to other paratenic or terminal hosts [31]. A large amount of research has confirmed that the liver is the preferred site for *T. canis* larvae. Maruyama et al. infected White Leghorn with 1500 eggs and found larvae (40–192) in liver tissues at 1, 3, 6, 10, 30, and 50 days post-infection (dpi), with the highest number of larvae collected at 1 dpi [32]. This suggests that the larvae could parasitize the liver for an extended period of time. In addition, larvae could occasionally reach other sites for parasitism, for example, muscle tissue, heart, spleen, and brain. Gargili et al. [33] further analyzed the migration of *T. canis* in chicks by artificially infecting 15-day-old chicks with 5000 eggs. The larvae mainly parasitized the liver (92.87% to 99.83%), followed by the lung, and occasionally migrated to the brain from 2 dpi to 12 dpi. But they did not detect larvae in the muscle tissue of the chicks. In contrast to the findings of previous experiments, Taira et al. confirmed that the larvae of *T. canis* could carry out hepatopulmonary migration in chickens [34]. However, the specific mechanism of migration is unknown. In summary, chickens of different ages and breeds are susceptible to *T. canis*, and the larvae can migrate to various tissues and organs, but the liver and lung are their preferred parasitic sites.

### 2.2. Pigeon

The pigeon is a bird with rich nutritional value. According to market research data, China’s annual consumption of meat pigeons is more than 5 billion. Pigeon liver, heart, and other organs are rich in protein and vitamin A. Although there is no scientific evidence to support the claim that eating raw pigeon heart can treat epilepsy, some individuals still follow this traditional belief and include raw pigeon viscera in their diet. Studies have shown that pigeons can also be a paratenic host for *T. canis*. As early as 1964, Galvin et al. described the parasitism of *T. canis* larvae in pigeons. The results showed that the larvae parasitized mainly in the liver of pigeons, followed by the lungs. Other organs, such as the brain and muscles, have few or no larvae [35]. In addition, Rahbar et al. further found that the average detection rate of larvae in tissues of pigeons at 1 dpi was significantly lower than that at 2, 3, 4, and 30 dpi, and the larval detection rate was the highest (55%) at 3 dpi [36]. In summary, the liver is still the preferred site of *T. canis* parasitism in pigeons, and the infected larvae penetrate the intestinal wall and migrate to various organs after 24 hpi. After 3 dpi, the parasitic larvae in the tissues begin to die.

### 2.3. Quail

The quail is a small bird, which is loved by people for its tender meat and rich nutritional value. Consuming raw quail eggs has been known to potentially mitigate skin allergies, rubella patches, and vomiting that can be triggered by the consumption of fish and shrimp. It may also help alleviate allergies resulting from specific medication injections [37]. Occasionally, raw consumption of its offal is also practised. As a bird, quail can also be a paratenic host for *T. canis*. Pahari and Sasmal conducted a study on the migration and distribution of *T. canis* larvae in Japanese quails [38]. A large number of larvae (92–95%) were collected from the liver of Japanese quail at 4, 10, and 20 dpi, and only a few larvae migrated to other tissues such as lung, heart, muscle, and brain. At the same time, the 50 larvae collected from the livers of Japanese quails were orally infected in mice, and these mice were successfully infected. Then, they also confirmed the infectivity of *T. canis* larvae in earthworms [39]. This suggested that birds could cause infection not only by feeding on eggs in the soil but also by feeding on earthworms that contain infected larvae. Nakamura et al. further analyzed the effects of different infection doses on Japanese quail. Larvae can be detected in the liver of Japanese quail which is infected with 1500 eggs after 12 hpi and can still be detected in the liver after 200 dpi. The larvae were detected in the liver within 6 h after oral infection of 4000 or 15,000 eggs [40,41]. This indicated that with the increase in infection intensity, the migration time of larvae from the intestine to the tissue would be significantly shortened.

### 2.4. Pig

Pigs are integral to the global economy, food production, and as providers of essential sustenance. Researchers observed intestinal wall lesions in pigs infected with *T. canis* eggs [42] and collected a large number of larvae from lymph nodes around the small intestine, lymph nodes around the large intestine, lungs, and liver [43]. Consistent with the experimental results of Sommerfelt et al. [44], they found that the migration of the larvae began at the lymph nodes of the small intestine, which seemed to be the first barrier for the larvae to invade, and the larvae were transferred from the first week to the fourth week post-infection (wpi). In addition, the liver was the organ with the highest number of larvae found in the first week, but the number of parasites gradually decreased. Larvae in the lungs appear from 1 wpi to 3 wpi, peaking at 2 wpi. This result is consistent with the results of Done and Helwigh [42,43]. Through primary and challenge infection experiments, Taira et al. confirmed that the larvae of *T. canis* mainly parasitized the lungs of pigs, and the larval load in the lungs reached its peak at 14 dpi [45]. Although the larval density decreased significantly over time, larvae could still be detected from the lungs until 49 dpi. The most important result is that a small number of larvae were found in the muscle, brain, and even in the eyes. Larvae have a special affinity for the liver, and liver which contained larvae remains infectious to mice [46]. These results suggested that *T. canis* larvae could migrate and survive in pig tissues for more than one month, but parasitism of larvae in specific preferred sites did not improve in survival time, and most larvae appeared to be eliminated by the host in the early infection stage, although a few larvae appeared to migrate randomly to other tissues, such as muscle or brain. Thus, pigs may be relatively less harmful as paratenic hosts for *T. canis* than mice or poultry, in which the larvae may persist for more than a year [40,47].

### 2.5. Sheep

Sheep have a significant impact on food supply, economic development, culture, and environment. Lamb is a sought-after meat and is widely consumed across various regions. Additionally, sheep grazing assists in managing weed growth and curbing the spread of invasive species in grasslands and other ecosystems. Previous studies have found that the migration paths of *T. canis* larvae in sheep are quite similar to those in other paratenic hosts [48,49]. The liver and lungs of sheep are the most affected organs, and the ileum is the site where migrating larvae penetrate the intestinal wall. In addition, although larvae migrate from intestinal tissue within 24 hpi [50], the antigens of this parasite can persist for a long time. Furthermore, even after seven months of post-experimental infection, larvae were still detected alive in certain affected tissues, such as livers and lungs [51].

## 3. Epidemiological Investigation

### 3.1. Chicken

Between May 1975 and May 1976, Lee et al. conducted a questionnaire survey to assess the eating habits of 1048 residents (558 men and 490 women) residing in five regions across three cities of two provinces in Korea [52]. This study aimed to analyze the potential for visceral larval migration resulting from the consumption of raw livers from poultry and livestock. Among the 1048 residents, the rate of raw poultry liver eating was 5.9%. Although nematode larvae were not detected in 120 chicken liver samples, the possibility of parasitic infection from eating raw poultry liver should not be ignored. The results of epidemiological investigation showed that 58.5% of free-range chickens in Espirito Santo State of Brazil were infected with *T. canis* and 12.7% could be considered as true infected [27]. At the same time, a serological investigation of semi-free-range broilers in 15 small farms in northeast Brazil showed that the overall prevalence was as high as 93.3%. In addition, 22.9% of soil that was tested in 48 selected areas next to fences in semi-free-range areas was contaminated with *T. canis* eggs. Even more concerning was the fact that in all the small farms where free-range chicken serum was collected, dogs and cats were found to roam freely in the same areas where chickens were raised. Anti-*T. canis* antibodies’ positive rates among chicken samples from small farms (78.6%), backyards (57.0%), and slaughterhouses (42.8%) were significantly different. In addition, the detection rate of different methods is obviously different. A percentage of 89.9% of poultry sera detected the IgY antibody against *T. canis* in four markets in Brazil, but no larvae were detected in tissues that used the pepsin digestion method [28]. Such a high serological positive rate might be related to the possibility of cross-reactivity with other worms. However, this still proves to a certain extent that there are significant differences in the specificity and sensitivity of different detection methods. In southern Brazil, the serum positive rate for antibodies against *T. canis* in broilers raised in seven semi-intensive systems was up to 67.7% [53]. And the positive animal detection rate of each farm ranged from 29.6% to 100%.

Apart from serological detection methods, PCR is also a commonly used method, which has higher sensitivity and specificity. Zibaei et al. showed that 12.7% newly hatched Cobb chicks in Iran were infected with *T. canis* through amplifying the ITS-1 and ITS-2 fragments by PCR [54]. In the Zanjan province of Iran, 10.5% of chickens from traditional farms were found to be infected with *Toxocara* sp., while 57.1% tested positive for *T. canis* [55]. Dogs roam freely on the farm in some areas of Iran [56]. Therefore, the high percentage of infected poultry detected in farms may be due to inadequate protection and poor hygiene standards in coop, which may lead to ingestion of eggs excreted by dogs.

### 3.2. Pig

An article published in 1979 documented a serological survey on *T. canis* antibodies and larvae antigens in pigs in farms across the United Kingdom. The results showed that 4.5% of the pigs tested positive for anti-*T. canis* antibodies [57]. Due to the limited detection methods available at that time, the detection rate may have been lower than the actual situation.

### 3.3. Sheep

Through detecting the level of anti-*T. canis* antibodies in 400 sheep serum samples from Powys and Gwent, Lloyd found that the seropositive rate increased with increase in age [58]. The same results were obtained from serum samples of 365 sheep of different breeds and ages from a slaughterhouse and farm in Presidente Prudente, south-eastern Brazil. This may be due to repeated exposure to constant irritation from eggs or live larvae with age, leading to an increase in prevalence. Meanwhile, the positive rate of females was higher than that of males, which may be due to the direct contact of females with the pasture and the fact that males are confined to small areas to avoid attacking each other [59]. In addition, Rio Grande do Sul, as a major sheep-producing and sheep-consuming state in southern Brazil, had a total positive rate of 29.0% in serum samples of 1642 sheep from 95 farms in 21 municipalities. Thessaly is one of the largest sheep- and goat-producing regions in Greece, where 42.9% of sheep and 10.1% of goats tested positive for anti-*T. canis* antibodies [60]. The difference in risk could be attributed to the different diets and preferences. Sheep graze on mostly grass, whereas goats prefer to browse and eat weeds, brush, and tree bark. Consequently, sheep had a higher probability of encountering soil that was contaminated with infected eggs. Additionally, studies have shown that farmers often handle carcasses and offal improperly, leaving the carcasses of deceased animals unattended and feeding raw offal to dogs. This increases the likelihood of dogs being infected for the first time or reinfected. Although 29.5% of sheep were provided with supplementary feed, most of them were still grazed in pastures, so the seropositive rate was also positively correlated with the presence of stray and wild dogs [61].

In general, only a few countries have conducted epidemiological investigations of the prevalence of *T. canis* in food-producing animals. Although there are few available data, the high rate of infection warrants increased attention.

## 4. Case Report

The clinical symptoms of human toxocariasis patients vary depending on the location of the parasite. However, these symptoms are not specific to this parasitic disease and can resemble other illnesses, making the diagnosis of VLM patients challenging [62]. As a result, there have been relatively few documented cases of human toxocariasis. This review mainly emphasizes cases caused by the consumption of raw or undercooked meat or internal organs from paratenic hosts.

### 4.1. Chicken

In certain areas of Japan, raw chicken meat and chicken liver are traditional foods. Due to their unique dietary habits, most reported cases of humans being infected with *T. canis* after consuming raw or undercooked infected chicken have occurred in Japan. In 1988, a pair of 22-year-old Japanese twin brothers were infected with *Toxocara* sp. after consuming fresh raw liver slices and raw chicken giblets at a snack shop, but the specific species of *Toxocara* was not identified [6]. A familial case of visceral larval migration (VLM) caused by *T. canis* larva infection was reported by Morimatsu et al. [21]. Patient 1 was a 45-year-old Japanese man, and Patient 2 was a 71-year-old Japanese man, the father of Patient 1. The two patients were diagnosed with VLM by testing positive for an excretion-secretory antigen of *T. canis* larvae in serum and bronchoalveolar lavage fluid (BALF). At the same time, the larvae of *T. canis* were detected in chicken products from the same source consumed by both patients. This is also the first time that antibodies have been reported in BALF in VLM patients. In addition, a 36-year-old Japanese man with a raw chicken diet was diagnosed with pulmonary toxocariasis [63], and a 34-year-old Japanese man who had consumed seared chicken sashimi several times was also diagnosed with toxocariasis [64].

### 4.2. Duck

At present, there is only one reported case of toxocariasis by eating duck liver. In 2006, Charite University Hospital in Berlin, Germany, diagnosed a 55-year-old woman with toxocariasis, possibly caused by eating duck liver. The patient had sudden hemiplegia in the right leg, and serological tests showed a 30% increase in eosinophils and a significant increase in serum total IgE level. Magnetic resonance imaging revealed multiple lesions in the brain. The serum and cerebrospinal fluid were tested for anti-*T. canis* antibody, and the results showed a high positive. The husband of the patient only ate roast duck meat and had no clinical symptoms, but his serum antibody against *T. canis* was also positive [22].

### 4.3. Pig

The study focused on the students of the Bunong Aborigines, which is the fourth largest indigenous tribe in Taiwan. These students reside in the mountainous regions of Hei-dong District and Yan-ping District, located in Taitung County in eastern Taiwan. Their community is situated at an elevation of approximately 300–400 m above sea level. The results of epidemiological investigation showed that the overall seropositivity of the indigenous populations was significantly higher than that of the Han population [65]. And age, but not sex, seemed to be more associated with seropositivity. Most importantly, for aboriginal adults with a history of eating pig liver, this diet may lead to a high rate of *T. canis* infection.

### 4.4. Beef

Kimmig et al. performed serological antibody tests on healthy blood donors in Stuttgart and found a significant increase in infection rates among cattle breeders compared to low-risk groups [66]. In 1983, Vortel et al. reported the presence of *T. canis* larvae in liver biopsies for the first time [67]. The patient was a 40-year-old cattle feeder from Czechoslovakia who was admitted for gastrectomy after suffering from dyspepsia, leukocytosis, and eosinophilia for about four years. Tests of liver tissue samples taken during the operation revealed *T. canis* infection, but the source of the infection was not traced back. Raw beef liver is one of the most popular dishes in Korea, and some people believe that raw liver or raw meat is good for health. Since members of the same family may have the same dietary habits, outbreaks of foodborne parasitic diseases tend to occur within the same family. Yoshikawa et al. reported three adult cases of visceral toxocariasis from the same family caused by weekly consumption of raw beef liver flakes [68]. Results of an epidemiological survey in Korea showed that 79.1% of 86 adults who were serologically positive for *T. canis* had a recent history of ingestion of raw beef liver or meat [69]. However, the study population may not be representative of the general population because the survey randomly selected those subjects who volunteered to answer. The data can still elucidate that eating raw beef liver or meat is still one of the most important risk factors for infection. Because the larva of *T. canis* occasionally invades the nervous system, the infection may cause some neurological symptoms. Recent epidemiological studies and meta-analyses have revealed a significant association between *Toxocara* infection and a range of debilitating neurological conditions, including cognitive impairment, psychosis, and epilepsy [70,71,72]. Additionally, there is emerging evidence suggesting a potential connection between *Toxocara* infection and the development of degenerative conditions like Alzheimer’s disease [73]. A 59-year-old male who was infected with *T. canis* by eating raw beef liver developed myelitis [74]. Similarly, a 21-year-old Japanese woman who was habituated to eating raw beef meat and liver developed VLM, and *T. canis* larvae had invaded the central nervous system [75]. Kambe et al. reported the first case of toxocariasis with neuromyelitis optica spectrum disorders (NMOSD) in a 53-year-old woman after consuming raw beef liver for one month [76]. Toxocariasis sometimes presents as myelitis or optic neuritis [15], and both diseases are characteristic of NMOSD. Therefore, in cases of NMOSD combined with toxocariasis, the effect of toxocariasis infection on neurological symptoms may be overlooked.

### 4.5. Sheep

There has been only one reported case of toxocariasis in a 63-year-old man from eating raw sheep liver [23]. The patient developed abdominal pain, diarrhea, fever, and other symptoms 2 h after eating raw sheep liver, and lung symptoms appeared two days later. These symptoms lasted for two weeks. Serological test results showed a significant increase in white blood cells and eosinophils, and ELISA test results confirmed that he had toxocariasis.

## 5. Conclusions and Future Directions

In summary, experimental infection results, epidemiological investigation data, and case reports have confirmed that a variety of food-producing animals can be paratenic hosts for *T. canis*. Due to the raw food diet in some countries and regions, animal-derived foods have become potential risk factors for human toxocariasis. However, the relevant epidemiological data are still relatively few, mainly due to the lack of attention to the infection of *T. canis* in its paratenic host, and the absence of accurate, sensitive, and easy-to-operate rapid diagnostic methods of toxocariasis. The current common diagnostic method is fecal flotation, which is time-consuming and labor-intensive, requires professional diagnosis, and requires molecular biological assays for further species identification. The absence of rapid diagnostic methods makes it challenging for researchers to conduct comprehensive epidemiological surveys, ultimately hindering the acquisition of accurate data of the infection rate. Consequently, the absence of reports about the infection in birds or mammals in most countries does not necessarily indicate the absence of toxocariasis within the region; rather, it may be a result of inadequate investigation. Healy et al. emphasized the importance of rigorous control and testing throughout every stage of the production chain, beginning with the farm and ending with the consumer [20]. To illustrate, it is essential to maintain the cleanliness of the breeding site to prevent infestation by infective eggs. Regular fecal testing should be conducted on the food-producing animals to detect any infections. Within the breeding site, the free movement of dogs should be prohibited. Furthermore, during the slaughtering process, the detection of parasite infection must be carried out. For consumers themselves, it is crucial to ensure that meat is fully cooked, and it is advisable to avoid consuming raw or undercooked meat products. Therefore, it is more important that we not only need to develop a rapid diagnostic method for toxocariasis, but also need to develop accurate diagnostic methods for each step of food production. Of course, a more economical and effective method is to develop a multi-faceted approach to address different steps in food production, thereby assessing the flow of *T. canis* from the environment to animals to animal-derived foods. In conclusion, it is crucial to prioritize the investigation of *T. canis* infection in food-producing animals, even in regions where no cases have been reported, to ensure comprehensive coverage and accurate assessment of the situation. Furthermore, the development of rapid diagnostic methods is imperative, as they can significantly enhance epidemiological investigations and provide crucial support for the prevention and control of toxocariasis.

In addition, *Toxocara cati* is a common zoonotic parasite found in cats. Similar to *T. canis*, the infective larval stage of *T. cati* is capable of migrating through the tissues of various paratenic hosts, such as mammals and birds. As a result, *T. cati* can also cause toxocariasis in humans. Sierra et al. [77] conducted experimental infection studies and provided evidence that pigs can serve as paratenic hosts for *T. cati*. Therefore, it is crucial to not only focus on the infection of *T. canis* in food-producing animals, but also to consider the infection risk of *T. cati* in these animals.

## Data Availability

No new data were created or analyzed in this study. Data sharing is not applicable to this article.

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
