# Peer review of "Prevalence, Infection, and Risk to Human Beings of Toxocara canis in Domestic Food-Producing Animals"

_vetsci, 2024, doi:10.3390/vetsci11020083_

Round 1

Reviewer 1 Report

Comments and Suggestions for Authors

The review on the epidemiologicl impact of toxocariasis by Toxocara canis in food producing animals, with zoonotic consequences is of interest , the paper is well written and only minor question occurred

lines 86, 98 - please, delete "kind of"

line 144 - ,  please, specify the tissues, extending the sentence

line 263 - italics

Considering that Toxocara cati is another zoonotic parasite, the Authors should mention doi: 10.1016/j.exppara.2020.107997; doi.org/10.1016/j.pt.2021.12.006 in conclusions

Author Response

  1. lines 86, 98 - please, delete "kind of"

Response: Thank you very much for your valuable advice. I have deleted it (Line 113, 129).

  1. line 144 -, please, specify the tissues, extending the sentence

Response: Thank you very much for your valuable advice. I have extended the sentence (Line 188-189).

  1. line 263 - italics

Response: Thank you very much for your valuable advice. I have changed it (Line 327, 330).

  1. Considering that Toxocara catiis another zoonotic parasite, the Authors should mention doi: 10.1016/j.exppara.2020.107997; doi.org/10.1016/j.pt.2021.12.006 in conclusions

Response: Thank you very much for your valuable advice. I have cited the articles in conclusion (Line 281-387).

Reviewer 2 Report

Comments and Suggestions for Authors

The authors have reviewed the literature on existence and sources of toxocariasis during several decades. Another review exists, published in 2017 and it should be cited in the references: Machado ER, de Araujo LB, de Leão e Neves Eduardo AM (2017) Human Toxocariasis: Secondary Data Analysis. Ann Clin Cytol Pathol 3(6): 1075.

The present review focuses on food-borne origin of the disease in humans, based on experimental and natural infection of production animals such as birds, pigs and cattle source of food for humans. The description of production animals harbouring infection is the core of the authors review and is new and interesting. The description of the disease or treatment in humans could be reduced since the review of Machado et al. was well documented in this area.

The authors insisted on food-borne origin, but infection in humans can also arise from contact with dogs/cats (one study in migrating populations in Pakistan), contaminated water, or vegetables. It could also arise from contact with contaminated sand and parks, since in several areas, the children between 2 and 5 years of age are the largely most infected.  Apparently, the ingestion of raw liver is a practice in Korea or Japan and possibly Taiwan, and then the source of infection is clearly from food. I think the introduction should propose those two ways of infection: direct (cats, dogs), nearly direct (sand of playing gounds for children) or indirect from paratenic hosts used for food.

The title mention T. canis but T. cati seems to be as frequent in Iranian chicken (paper of Brazilian authors from 2022), so the title could include Toxocara sp.

There is one mistake in the citation of Dubey et al: lines 58-59: The paper is on Toxoplasmosis and the prevalences are of Toxoplasma and not Toxocara.

Author Response

  1. The authors have reviewed the literature on existence and sources of toxocariasis during several decades. Another review exists, published in 2017 and it should be cited in the references: Machado ER, de Araujo LB, de Leão e Neves Eduardo AM (2017) Human Toxocariasis: Secondary Data Analysis. Ann Clin Cytol Pathol 3(6): 1075.

Response: Thank you very much for your valuable advice. I have cited the references in our manuscript (Line 263-268).

  1. The present review focuses on food-borne origin of the disease in humans, based on experimental and natural infection of production animals such as birds, pigs and cattle source of food for humans. The description of production animals harbouring infection is the core of the authors review and is new and interesting. The description of the disease or treatment in humans could be reduced since the review of Machado et al. was well documented in this area.

Response: Thank you very much for your valuable advice. Our primary focus was on the cases of T. canis infection that occur when contaminated paratenic host meat is consumed. We placed more emphasis on the cause of the infection and provided only a brief overview of the symptoms. In contrast, Machado et al. provided a detailed description of the various clinical manifestations of human toxocariasis.

  1. The authors insisted on food-borne origin, but infection in humans can also arise from contact with dogs/cats (one study in migrating populations in Pakistan), contaminated water, or vegetables. It could also arise from contact with contaminated sand and parks, since in several areas, the children between 2 and 5 years of age are the largely most infected.  Apparently, the ingestion of raw liver is a practice in Korea or Japan and possibly Taiwan, and then the source of infection is clearly from food. I think the introduction should propose those two ways of infection: direct (cats, dogs), nearly direct (sand of playing gounds for children) or indirect from paratenic hosts used for food.

Response: Thank you very much for your valuable advice. Infections can indeed occur through direct contact with dogs or exposure to contaminated soil. While we do not dispute this, we emphasize that food producing animal-borne infections are often overlooked. To ensure greater accuracy in our expression, we have included additional relevant descriptions in the introduction section (Line 34-40).

  1. The title mention canisbut T. cati seems to be as frequent in Iranian chicken (paper of Brazilian authors from 2022), so the title could include Toxocara sp.

Response: Thank you very much for your valuable advice. This study primarily focused on the infection of T. canis in food producing animals. However, due to the limited availability of early investigation data, the species was not always accurately identified. Therefore, some of the epidemiological data presented in this study may be specific to Toxocara sp. in general, rather than T. canis specifically. Furthermore, if the discussion of T. catis was to be included, the article would become even more extensive.

  1. There is one mistake in the citation of Dubey et al: lines 58-59: The paper is on Toxoplasmosis and the prevalences are of Toxoplasma and not Toxocara.

Response: Thank you very much for your valuable advice. We have deleted the content.

Reviewer 3 Report

Comments and Suggestions for Authors

Dear author,

The manuscript title is “Prevalence, infection and risk to human beings of Toxocara canis in domestic food-producing animals and it aims to review the occurrence of Toxocara canis infection in farmed animals.

The topic falls within the aims and scope of the journal. It is relevant in the field of zoonosis.

Many references are very very old, many have between 40 – 50 years! They could only be used as historical references and they are especially strange when mixed with recent ones on PCR and so on.

Some particular suggestions/comments will be done here:

-       Abstract: an English review is needed; it is not said what methodology was used to perform this review; inclusion/exclusion factors, keywords used, among others

-       Lines 15 – I suggest the authors not to repeat words in keywords that are already in the title

-       Lines 25 – 26: there are several pathogens that survive at 4ºC, that is not an immediate demonstration of the zoonotic potential, as the problem is not to cook the food

-       Line 54 – There is no methodology. The authors must explain how did they perform the revision, how did they collect papers, in which libraries, inclusion/exclusion factors and other parameters needed to perform a review.

-       Line 56 – Traditionally?? Where?? Mostly broilers are raised in super-intensive farms.

-       Lines 238 – 240 and elsewhere – the authors should detail the localization of the studies, because indigenous population may be from several different countries

-       Lines 262 – 266 – please add italics when needed (taxonomy)

-       Line 291 – absence? It is possible to do PCR and serology for Toxocara diagnose

-       Some references are from journals that are not indexed and are mostly unknown

Author Response

  1. Abstract: an English review is needed; it is not said what methodology was used to perform this review; inclusion/exclusion factors, keywords used, among others

Response: Thank you very much for your valuable advice. I have added the methodology used in abstract (Line 11-13).

  1. Lines 15 – I suggest the authors not to repeat words in keywords that are already in the title

Response: Thank you very much for your valuable advice. I have changed the keywords.

  1. Lines 25 – 26: there are several pathogens that survive at 4ºC, that is not an immediate demonstration of the zoonotic potential, as the problem is not to cook the food

Response: Thank you very much for your valuable advice. I have deleted the sentence.

  1. Line 54 – There is no methodology. The authors must explain how did they perform the revision, how did they collect papers, in which libraries, inclusion/exclusion factors and other parameters needed to perform a review.

Response: Thank you very much for your valuable advice. I have added the methodology we utilized in the introduction section (Line 68-78).

  1. Line 56 – Traditionally?? Where?? Mostly broilers are raised in super-intensive farms.

Response: Thank you very much for your valuable advice. I have rewritten the sentence (Line 82).

  1. Lines 238 – 240 and elsewhere – the authors should detail the localization of the studies, because indigenous population may be from several different countries

Response: Thank you very much for your valuable advice. We describe the indigenous population in detail (Line 297-301).

  1. Lines 262 – 266 – please add italics when needed (taxonomy)

Response: Thank you very much for your valuable advice. I have changed them in italics.

  1. Line 291 – absence? It is possible to do PCR and serology for Toxocara diagnose

Response: Thank you very much for your valuable advice. I have rewritten the sentence (Line 337-338).

  1. Some references are from journals that are not indexed and are mostly unknown

Response: Thank you very much for your valuable advice. I have changed some references.

Reviewer 4 Report

Comments and Suggestions for Authors

Introduction. This chapter provides a comprehensive introduction to the topic of toxocariasis, highlighting its global prevalence, zoonotic nature, and neglected status. Here are some positive aspects of the chapter:

¾    The chapter clearly establishes the problem of toxocariasis as a prevalent and neglected zoonotic disease, emphasizing its significance on a global scale. This sets the stage for the reader to understand the importance of further investigation.

¾    The text recognizes the higher prevalence of toxocariasis in developing countries, especially in regions with diets involving uncooked or raw meat. This global perspective helps frame the issue in a broader context.

¾    The chapter effectively discusses the soil-borne nature of toxocariasis and its transmission through the consumption of raw or undercooked meat and internal organs from paratenic hosts. This provides a clear understanding of how humans can contract the disease.

¾    The section discussing the potential consequences of toxocariasis in humans, including severe complications in the liver, lung, eye, and nervous system, adds a layer of significance to the topic.

¾    The inclusion of epidemiological data, such as variations in serum prevalence among different populations and the increased risk for certain groups, provides a quantitative aspect to the narrative.

¾    Mentioning reported cases of human toxocariasis due to the consumption of various animal livers adds concrete examples to the discussion, making it more relatable.

¾    The decision to focus the review on food-producing animals infected with T. canis contributes to a more targeted exploration of potential risk factors for human toxocariasis.

¾    Call to Action: The chapter concludes with a clear purpose – to review and summarize the parasitism, migration, and infection of T. canis in food-producing animals. The goal is to elucidate potential risk factors for human toxocariasis, leading to the development of prevention and control measures.

However, a potential improvement could be providing more recent statistics or updates on the prevalence of toxocariasis, if available. Additionally, clarity on the specific focus and objectives of the review could enhance the reader's understanding. Overall, the chapter lays a solid foundation for exploring the different facets of toxocariasis and its potential impact on human health.

Experimental infection

1.1. Chicken

What is the traditional rearing system for broilers, and how does it contribute to their susceptibility to soil-borne parasites?

What is the significance of Dubey et al.'s findings regarding the detection rate of parasites in free-range chickens, especially in relation to Toxocara spp.?

How does the experimental inoculation of T. canis eggs into the gizzard of chickens, as conducted by Inoue et al., demonstrate the potential of chickens as paratenic hosts?

What evidence supports the claim that chickens can release T. canis eggs into the environment post-infection, and what are the implications for cross-infection and increased risk for breeders?

How do studies confirm the preferred parasitic sites of T. canis larvae in chickens, especially regarding their migration patterns and the role of the liver?

1.2. Pigeon

What is the nutritional value of pigeons, and how does their consumption relate to their role as paratenic hosts for T. canis?

How did Galvin et al. describe the migration pattern of T. canis larvae in pigeons, and what organs are primarily affected?

What are the key findings regarding the detection rates and survival of T. canis larvae in pigeons' tissues at different post-infection time points?

How does the information about pigeons as paratenic hosts compare with that of chickens in terms of preferred parasitic sites and migration patterns of T. canis larvae?

1.3. Quail

How does the nutritional value and popularity of quail relate to their potential role as paratenic hosts for T. canis?

What evidence supports the claim that quail can be paratenic hosts for T. canis, and what organs are primarily affected based on the study by Pahari and Sasmal?

How does the study by Nakamura et al. contribute to our understanding of the effects of different infection doses on Japanese quail and the migration of T. canis larvae in quail tissues?

1.4. Pig

How do pigs, as food-producing animals, impact the world economy, food supply, and human health?

What evidence supports the claim that pigs can be paratenic hosts for T. canis, and what are the preferred parasitic sites and migration patterns of T. canis larvae in pigs?

How does the study by Taira et al. provide insights into the parasitism of T. canis larvae in pigs, including the organs affected and the duration of larval survival in pig tissues?

1.5. Sheep

What is the significance of sheep in terms of food supply, economic development, culture, and environment?

How do the migration paths of T. canis larvae in sheep compare to those in other paratenic hosts, especially regarding the affected organs and the penetration of the intestinal wall?

What evidence suggests the persistence of T. canis antigens in sheep tissues, and how does the longevity of larvae in certain affected tissues impact our understanding of sheep as paratenic hosts?

2. Epidemiological investigation

¾    Do you have access to more recent data or studies on the epidemiology of T. canis in food-producing animals? An update on recent findings would enhance the relevance of the information.

¾    Interpretation of Seropositivity: How does the chapter interpret the seropositivity rates in different studies? Are there any variations in the interpretation of serological results, and if so, how are these variations addressed?

¾    Differences in Grazing Behaviors: The chapter mentions differences in grazing behaviors affecting the prevalence of T. canis. Could you elaborate on how specific grazing behaviors influence the risk of infection in sheep and other animals?

¾    How do management practices, such as the handling of carcasses and offal, influence the prevalence of T. canis in sheep and other food-producing animals? Is there evidence suggesting that certain practices contribute to higher infection rates?

¾    Does the chapter identify any specific gaps in the existing epidemiological research on T. canis in food-producing animals? Are there areas where further investigation is particularly needed?

Overall, the chapter provides a solid foundation in understanding the epidemiology of T. canis in food-producing animals, but considering the limitations of historical data, updating with recent findings and addressing specific research gaps would enhance its completeness.

3. Case report

Questions:

¾    Are there notable differences in the reported cases and prevalence of T. canis infection in different regions or countries? How does the global distribution of these cases contribute to the understanding of the disease?

¾    Could the chapter elaborate on why certain dietary habits, such as the consumption of raw beef liver, are identified as potential risk factors for T. canis infection? Are there specific practices or cultural reasons contributing to these dietary choices?

¾    How do the reported cases and epidemiological data discussed in this chapter contribute to public health considerations and awareness regarding the risks associated with consuming specific animal products?

¾    Does the chapter discuss any preventive measures or recommendations to reduce the risk of T. canis infection from consuming raw or undercooked meat? How can public awareness be raised to minimize such risks?

¾    Given the advancements in diagnostic techniques and changes in dietary habits, are there more recent cases or studies that provide additional insights into T. canis infection from food-producing animals?

Overall, the chapter provides valuable insights into the diverse cases of T. canis infection associated with the consumption of various animal products. Addressing the questions above may further enhance the chapter's relevance and completeness.

Summary

The provided summary is concise and effectively captures the key points discussed in the chapter. It successfully conveys the main findings from experimental infections, epidemiological investigations, and case reports, emphasizing the role of various food-producing animals as paratenic hosts for T. canis. Additionally, the summary highlights the significance of raw food diets in certain regions and the potential risk factors for human toxocariasis.

However, to enhance the completeness of the summary, you may consider addressing the following points:

¾    Highlight the potential implications of these findings on public health, emphasizing the importance of raising awareness, implementing preventive measures, and developing strategies for the control of toxocariasis.

¾    Connect the experimental infection results, epidemiological data, and case reports to provide a more integrated overview of the evidence supporting the role of food-producing animals in the transmission of T. canis to humans.

¾    Consider adding a brief mention of potential future research directions, such as exploring novel diagnostic methods, investigating the impact of changing dietary habits, and identifying specific risk factors associated with different animal hosts.

Author Response

  1. What is the traditional rearing system for broilers, and how does it contribute to their susceptibility to soil-borne parasites?

Response: Thank you very much for your valuable advice. The traditional farming method of broiler chickens in rural areas involves free-range or semi-free-range systems, as these systems allow the chickens to roam freely, facilitating easier access to eggs in the soil (Line 80-84).

  1. What is the significance of Dubey et al.'s findings regarding the detection rate of parasites in free-range chickens, especially in relation to Toxocara spp.?

Response: Thank you very much for your valuable advice. The study conducted by Dubey et al. on the infection rate of T. gondii in chickens has been excluded.

  1. How does the experimental inoculation of caniseggs into the gizzard of chickens, as conducted by Inoue et al., demonstrate the potential of chickens as paratenic hosts?

Response: Thank you very much for your valuable advice. They confirmed through artificial infection experiments that larvae can migrate from the gizzard to the liver, thus proving that chickens can serve as paratenic hosts for T. canis.

  1. What evidence supports the claim that chickens can release canis eggs into the environment post-infection, and what are the implications for cross-infection and increased risk for breeders?

Response: Thank you very much for your valuable advice. Reference 30 describes this fact in detail (Line 86-89).

  1. How do studies confirm the preferred parasitic sites of canislarvae in chickens, especially regarding their migration patterns and the role of the liver?

Response: Thank you very much for your valuable advice. The mode of larval migration has not been fully explained, only the existence of the hepatopulmonary migration pathway has been demonstrated, and the reason for the migration to the liver is not understood. We only showed that the liver was the preferred site of the larval larvae by the large number and long-term parasitism in the liver.

  1. What is the nutritional value of pigeons, and how does their consumption relate to their role as paratenic hosts for canis?

Response: Thank you very much for your valuable advice. Pigeon liver, heart, and other organs are rich in protein and vitamin A. Although there is no scientific evidence to support the claim that eating raw pigeon heart can treat epilepsy, some individuals still follow this traditional belief and include raw pigeon viscera in their diet.

  1. How did Galvin et al. describe the migration pattern of canislarvae in pigeons, and what organs are primarily affected?

Response: Thank you very much for your valuable advice. Galvin et al. did not describe the migration pattern of larvae, and in order to be more accurate, it was changed to "parasitism” (Line 119).

  1. What are the key findings regarding the detection rates and survival of canis larvae in pigeons' tissues at different post-infection time points?

Response: Thank you very much for your valuable advice. The findings showed that larva mainly parasitized liver and lung and could survive for a long time.

  1. How does the information about pigeons as paratenic hosts compare with that of chickens in terms of preferred parasitic sites and migration patterns of canislarvae?

Response: Thank you very much for your valuable advice. Compared with chickens, there is no difference in the preference of parasitic sites of pigeons. Due to the lack of relevant literature, the migration mode of T. canis in pigeons is not known at present.

  1. How does the nutritional value and popularity of quail relate to their potential role as paratenic hosts for canis?

Response: Thank you very much for your valuable advice. Consuming raw quail eggs has been known to potentially mitigate skin allergies, rubella patches, and vomiting that can be triggered by the consumption of fish and shrimp. It may also help alleviate allergies resulting from specific medication injections.

  1. What evidence supports the claim that quail can be paratenic hosts for canis, and what organs are primarily affected based on the study by Pahari and Sasmal?

Response: Thank you very much for your valuable advice. A large number of larvae parasitism were found in the liver of quail, while only a small number of larvae parasitism were found in other tissues.

  1. How does the study by Nakamura et al. contribute to our understanding of the effects of different infection doses on Japanese quail and the migration of T. canis larvae in quail tissues?

Response: Thank you very much for your valuable advice. Nakamura et al. confirmed that the higher the dose of infection, the shorter the time required for larvae to migrate from the intestine to the liver.

  1. How do pigs, as food-producing animals, impact the world economy, food supply, and human health?

Response: Thank you very much for your valuable advice. I have added the content in Line 149-153.

  1. What evidence supports the claim that pigs can be paratenic hosts for T. canis, and what are the preferred parasitic sites and migration patterns of T. canis larvae in pigs?

Response: Thank you very much for your valuable advice. After artificial infection of eggs, a large number of larvae can be collected in porcine mesenteric lymph nodes, liver and lung. The preferred sites of parasitism are liver and lung. At present, only the mode of parasitism from intestinal tract to various tissues is known, and whether there is any migration between tissues is not known.

  1. How does the study by Taira et al. provide insights into the parasitism of T. canis larvae in pigs, including the organs affected and the duration of larval survival in pig tissues?

Response: Thank you very much for your valuable advice. Taira et al. conducted primary infection and secondary infection experiments, and extended the detection time to 49 days post infection.

  1. What is the significance of sheep in terms of food supply, economic development, culture, and environment?

Response: Thank you very much for your valuable advice. I have added the content in Line 177-182.

  1. How do the migration paths of canis larvae in sheep compare to those in other paratenic hosts, especially regarding the affected organs and the penetration of the intestinal wall?

Response: Thank you very much for your valuable advice. The migration pattern of T. canis in sheep is similar to that in other paratenic hosts, with the liver and lungs being the main parasitic sites and the ileum being the site where migrating larvae penetrate the intestinal wall.

  1. What evidence suggests the persistence of canisantigens in sheep tissues, and how does the longevity of larvae in certain affected tissues impact our understanding of sheep as paratenic hosts?

Response: Thank you very much for your valuable advice. Although the larvae migrate from the gut to the liver 24 hours after infection, antigens can be detected in the gut for a long time. And larvae could still be detected in infected tissue seven months after infection.

  1. Do you have access to more recent data or studies on the epidemiology of T. canis in food-producing animals? An update on recent findings would enhance the relevance of the information.

Response: Thank you very much for your valuable advice. We do not yet have epidemiological data on T. canis in food-producing animals, and there is a lack of relevant research reports in China. Our future research will focus on this aspect.

  1. Interpretation of Seropositivity: How does the chapter interpret the seropositivity rates in different studies? Are there any variations in the interpretation of serological results, and if so, how are these variations addressed?

Response: Thank you very much for your valuable advice. The serological detection methods used in different experiments are different, and the sensitivity and specificity of different detection methods are also different, so there are certain differences in the seropositivity of different studies.

  1. Differences in Grazing Behaviors: The chapter mentions differences in grazing behaviors affecting the prevalence of T. canis. Could you elaborate on how specific grazing behaviors influence the risk of infection in sheep and other animals?

Response: Thank you very much for your valuable advice. Sheep have a natural tendency to gather in groups and graze closely together, whereas goats are typically more spread out and prefer to stay farther apart. They also have different preferences when it comes to food. Sheep generally favor consuming the leaves and stems of plants, while goats prefer to eat the roots and fruits. Goats have more developed salivary glands that aid in digesting cellulose, allowing them to consume rough grass and thorns. When herding, goats tend to move at a faster pace compared to the relatively calm and steady movement of sheep. Additionally, studies have shown that farmers often handle carcasses and offal improperly, leaving the carcasses of deceased animals unattended and feeding raw offal to dogs.

  1. How do management practices, such as the handling of carcasses and offal, influence the prevalence of canisin sheep and other food-producing animals? Is there evidence suggesting that certain practices contribute to higher infection rates?

Response: Thank you very much for your valuable advice. Epidemiological results showed that farmers' management of carcasses and offal is generally inappropriate, i.e. carcasses of dead animals are abandoned and raw offal is fed to dogs. Feeding raw offal to dogs may lead to infection of dogs, infected dogs shed eggs with feces, and sheep may be infected by feeding during grazing.

  1. Does the chapter identify any specific gaps in the existing epidemiological research on T. canis in food-producing animals? Are there areas where further investigation is particularly needed?

Response: Thank you very much for your valuable advice. At present, there are relatively few studies, and only a few countries have reported relevant epidemiological data. The next step is to draw the attention of researchers to the infection situation, so as to carry out relevant epidemiological investigations in more countries and regions.

  1. Are there notable differences in the reported cases and prevalence of T. canis infection in different regions or countries? How does the global distribution of these cases contribute to the understanding of the disease?

Response: Thank you very much for your valuable advice. There is some difference between the number of reported cases and the infection rate, first of all, some people have a negative infection state, because there is no obvious clinical symptoms, they will not go to the hospital to detect. From the distribution of cases reported so far, the infection is mainly related to the dietary habits of this country or region.

  1. Could the chapter elaborate on why certain dietary habits, such as the consumption of raw beef liver, are identified as potential risk factors for T. canis infection? Are there specific practices or cultural reasons contributing to these dietary choices?

Response: Thank you very much for your valuable advice. We went into more detail about why certain dietary habits, such as consumption of raw beef liver, were identified as potential risk factors for T. canis infection. In Japanese culture, food is seen as a connection to nature and the universe. Eating raw liver is seen as a way to maintain its original taste and nutritional value, so it is much loved. In addition, eating raw liver is also related to traditional Japanese medical theories, which believe that it helps improve health and beauty.

  1. How do the reported cases and epidemiological data discussed in this chapter contribute to public health considerations and awareness regarding the risks associated with consuming specific animal products?

Response: Thank you very much for your valuable advice. Through epidemiological investigation results and case reports, people are aware of the possibility of infection caused by poor eating habits and the infection rate is higher than excepted, thus arousing public concern and attention.

  1. Does the chapter discuss any preventive measures or recommendations to reduce the risk of T. canis infection from consuming raw or undercooked meat? How can public awareness be raised to minimize such risks?

Response: Thank you very much for your valuable advice. In the discussion section, we put forward some relevant suggestions.

  1. Given the advancements in diagnostic techniques and changes in dietary habits, are there more recent cases or studies that provide additional insights into T. canis infection from food-producing animals?

Response: Thank you very much for your valuable advice. There are few epidemiological data and case reports, and no evidence to show how infections have changed recently with the development of diagnostic techniques and changes in people's diet.

  1. Highlight the potential implications of these findings on public health, emphasizing the importance of raising awareness, implementing preventive measures, and developing strategies for the control of toxocariasis.

Response: Thank you very much for your valuable advice. I have added these content in conclusion.

  1. Connect the experimental infection results, epidemiological data, and case reports to provide a more integrated overview of the evidence supporting the role of food-producing animals in the transmission of T. canis to humans.

Response: Thank you very much for your valuable advice. I have added these content in conclusion.

  1. Consider adding a brief mention of potential future research directions, such as exploring novel diagnostic methods, investigating the impact of changing dietary habits, and identifying specific risk factors associated with different animal hosts.

Response: Thank you very much for your valuable advice. I have added these content in conclusion.

Round 2

Reviewer 2 Report

Comments and Suggestions for Authors

The paper is now OK for me? although there are some minor modifications to be made.

l 236 : I propose " Goats and goats have different diets and preferences. Sheep graze on mostly grass, whereas goats prefer to browse and eat weeds, brush, tree bark, etc."instead of what was written on their feeding habits.

Try to relate these habits to the actual infection of these ruminants with Toxocara. Same comment for other productions. Those reared inside without contact with dogs have a reduced chance to become infected (industrial pigs, sheep reared indoors etc..).

For animal food productions, you have added some comments on their importance. You could shorten these comments when they are not related to Toxocara infection.

Author Response

  1. 236 : I propose " Goats and goats have different diets and preferences. Sheep graze on mostly grass, whereas goats prefer to browse and eat weeds, brush, tree bark, etc."instead of what was written on their feeding habits.

Response: Thank you very much for your valuable advice. I have rewritten this sentence (Line 243-246).

  1. Try to relate these habits to the actual infection of these ruminants with Toxocara. Same comment for other productions. Those reared inside without contact with dogs have a reduced chance to become infected (industrial pigs, sheep reared indoors etc..).

Response: Thank you very much for your valuable advice. I have added the contents.

  1. For animal food productions, you have added some comments on their importance. You could shorten these comments when they are not related to Toxocara infection.

Response: Thank you very much for your valuable advice. I have deleted some sentence.

Reviewer 3 Report

Comments and Suggestions for Authors

Dear authors,

Thank you for the improvements in your manuscript. Nevertheless, methodology is still weak.

Line 11 – a systematic review must comply with a series of rules that did not happen here (please see https://guides.lib.unc.edu/systematic-reviews/). Maybe the authors should call it another thing rather than a systematic review

Line 17 – the transmission is never by direct contact but by ingestion, please correct

Line 35 – please write “Toxocara ingected dogs” instead of “positive”

Line 61 – what is a “relevant” case study? How did the authors classified the relevance? That is the methodology missing.

Lines 69-70 – previously the authors wrote Chinese and English…contradictory information in the manuscript

Lines 73 – 74 – previously the authors wrote only case studies, again, contradictory information

Lines 74 – 75 – somewhere the authors should state how many papers they have collected first, before the review, and how many stayed in the end to write this paper

Lines 370 – 377 – it is Toxacara cati, please correct all

Author Response

  1. Line 11 – a systematic review must comply with a series of rules that did not happen here (please see https://guides.lib.unc.edu/systematic-reviews/). Maybe the authors should call it another thing rather than a systematic review

Response: Thank you very much for your valuable advice. I have corrected this mistake.

  1. Line 17 – the transmission is never by direct contact but by ingestion, please correct

Response: Thank you very much for your valuable advice. I have corrected this mistake.

  1. Line 35 – please write “Toxocara infected dogs” instead of “positive”

Response: Thank you very much for your valuable advice. I have changed it (Line 35).

  1. Line 61 – what is a “relevant” case study? How did the authors classified the relevance? That is the methodology missing.

Response: Thank you very much for your valuable advice. I have described it in detail (Line 63-66).

  1. Lines 69-70 – previously the authors wrote Chinese and English…contradictory information in the manuscript

Response: Thank you very much for your valuable advice. I have corrected this mistake in the abstract (Line 11-14).

  1. Lines 73 – 74 – previously the authors wrote only case studies, again, contradictory information

Response: Thank you very much for your valuable advice. I have not only conducted case analysis, and have modified the previous contradictory information.

  1. Lines 74 – 75 – somewhere the authors should state how many papers they have collected first, before the review, and how many stayed in the end to write this paper

Response: Thank you very much for your valuable advice. I have added the content (Line 77-78).

  1. Lines 370 – 377 – it is Toxacara cati, please correct all

Response: Thank you very much for your valuable advice. I have changed the mistake.

Round 3

Reviewer 3 Report

Comments and Suggestions for Authors

This reviewer suggest authors to prepare better methodology next manuscript.

All the best